# A Corpus Approach to Roman Law Based on Justinian's Digest

**Marton Ribary [1],\* and Barbara McGillivray [2,3]**

1   School of Law, University of Surrey, Guildford GU2 7XH, UK
2   Theoretical and Applied Linguistics, Faculty of Modern and Medieval Languages and Linguistics, University of Cambridge, Cambridge CB3 9DA, UK; bm517@cam.ac.uk
3   The Alan Turing Institute, London NW1 2DB, UK
*   Correspondence: m.ribary@surrey.ac.uk

**Abstract:** Traditional philological methods in Roman legal scholarship such as close reading and strict juristic reasoning have analysed law in extraordinary detail. Such methods, however, have paid less attention to the empirical characteristics of legal texts and occasionally projected an abstract framework onto the sources. The paper presents a series of computer-assisted methods to open new frontiers of inquiry. Using a Python coding environment, we have built a relational database of the Latin text of the *Digest*, a historical sourcebook of Roman law compiled under the order of Emperor Justinian in 533 CE. Subsequently, we investigated the structure of Roman law by automatically clustering the sections of the *Digest* according to their linguistic profile. Finally, we explored the characteristics of Roman legal language according to the principles and methods of computational distributional semantics. Our research has discovered an empirical structure of Roman law which arises from the sources themselves and complements the dominant scholarly assumption that Roman law rests on abstract structures. By building and comparing Latin word embeddings models, we were also able to detect a semantic split in words with general and legal sense. These investigations point to a practical focus in Roman law which is consistent with the view that ancient law schools were more interested in training lawyers for practice rather than in philosophical neatness.

**Keywords:** Roman law; Digest; computational linguistics; corpus linguistics; clustering; distributional semantics; word embeddings; Python; Latin; LatinISE

## 1. Introduction

*Iustitia est constans et perpetua voluntas ius suum cuique tribuens* [1] (p. 37)—"Justice is a persistent and perpetual will to provide each person with his right." (MR trans.) These are the opening words of Justinian's *Institutes* issued together with the *Digest* of Roman law in 533 CE. In this article, we supplement and potentially challenge this abstract understanding of Roman law as we shift focus from law as a theoretical system of ideas to law as an empirical collection of texts.

Generations of Romanists have argued that ancient Roman legal education was underpinned by rhetorical rather than philosophical ideals. In a 2014 article about "the origin of legal argumentation in Roman law", Philip Thomas presents "the warming-up of the hypothesis by Stroux and Viehweg that the methodology of legal argument in Roman law derived from rhetoric" [2] (p. 43). Johannes Stroux recognised rhetorical techniques in juristic problem-solving as well juristic speech [3], an idea which was celebrated by Salvatore Riccobono as putting an end to the obsession of discovering interpolations in juristic writings [4]. Similarly, as Thomas describes, Theodor Viehweg [5] "distinguishes between topical, problem-orientated argumentation as opposed to axiomatic, systematic-deductive legal reasoning" [2] (p. 46). The thought-provoking idea generated a fruitful discussion about the theoretical foundations

of (Roman) law and its perceived systematicity. In a similar manner, Max Kaser distinguishes between system-based (European) and case-law oriented (Anglo-Saxon) traditions [6]. René Brouwer relates "system" in the former "civil law" tradition to "substantive doctrine" which he traces back to the Byzantine codification of Roman law under Stoic philosophical influence. In contrast, he relates the "system" in the English "common law" tradition to "the functioning of the law" [7] (p. 45–46). As much as the distinction holds firm and true to "the two main Western legal traditions", our research suggests that both tendencies are present in Roman law. While Justinian's *Institutes* presents an abstract framework focusing on substantive doctrine, the *Digest* seems to be arranged according to a practical structure focusing on the functioning of the law.

Our computer-assisted analysis brings out this latter tendency of Roman law by staying close to the empirical evidence provided by Justinian's *Digest* as a text corpus. Our findings suggest that the structure and language of Roman law as it is presented in the *Digest* is shaped not so much by a pre-conceived system, or an abstract idea of justice, but by the concrete processes in which justice is exercised, administered and delivered. Following this practical focus, we organise legal texts based on their vocabulary into textually similar clusters using an unsupervised approach. Additionally, we analyse the relationship between general Latin language and the Latin vocabulary of legal texts, and we explore how some words come to acquire a specifically legal sense.

The principal source of our investigation is the *Digest*, a monumental historical sourcebook of Roman law created on the order of Emperor Justinian I and dated to 533 CE. Understanding the structure and language of classical Roman law has traditionally been based on the close reading of selected *Digest* passages to which the medieval Glossators, the 19th century Pandectists and modern Romanists applied their mastery of juristic reasoning. We propose another approach which supplements this largely theoretical inquiry with an empirical one which examines the structure and language of Roman law as a corpus of texts. Our assumption is that while the *Digest* does not constitute a system of Roman law, it accurately represents what Roman law is. This assumption leads us to one negative and one positive derivative assumption. On the one hand, we do not assume that Roman law as presented in the *Digest* has an inferential structure similar to a mathematical knowledge domain like Euclid's *Elements* [8]. On the other hand, we do assume that thanks to the terminological nature of Roman law and the comprehensiveness of the *Digest*, the semantic structure of the text corpus indicates an underlying conceptual structure. For this reason, while a corpus approach does not discover a hidden system, it is capable of grasping the empirical structure of Roman law and demonstrating the characteristics of its vocabulary.

Our study is divided into three main parts. First, we describe the processing steps we carried out on the raw text of the *Digest* to prepare the corpus for machine-assisted analysis (Section 2 "Corpus"). Based on the ancient 6th century CE manuscript of the *littera Florentina* [9], cumulative philological efforts have produced a standard printed edition [10] and one of the earliest digital text editions in the history of what we now call Digital Humanities. The revolutionary ROMTEXT was originally produced on punch cards in the 1970s, which were subsequently converted to a command line format in DOS [11]. We have produced a relational database from ROMTEXT [12], which can be seen as the natural next step in philological efforts spanning more than a millennium.

The second part of our analysis aimed at discovering an empirical structure of Roman law based on the *Digest* and using computational clustering methods (Section 3 "Clustering"). We applied hierarchical clustering onto the 432 thematic sections created by the ancient editors. As the *Digest* represents the full scope of Roman law in its classical form, hierarchical clustering revealed not just larger themes within the corpus, but it also indicated how these themes relate to each other in a hierarchical tree-like structure. The institutional framework of Roman law or the architectonic structure created by the Pandectists present Roman law according to preconceived philosophical ideas. In stark contrast to this abstract structure, hierarchical clustering reveals an empirical structure whose main organising principle is informed by law as practice. The finding is coherent with David Pugsley's fascinating but controversial theory about the genesis of the *Digest* [13], which he sees as reflecting

centuries of legal educational practice based primarily on the commentaries of the jurist Ulpian (ca. 170–228 CE). Those ancient textbooks excerpted from Ulpian's commentaries were designed to train lawyers for practice, not lawyers for a philosophical system. As our finding suggests, school practice as reflected in the *Digest* focused on exercising, administering and delivering law. The focus was on how one practices law—and from that perspective, it is largely irrelevant whether law holds neatly together in a system.

We notice a similar practical orientation as we focus our analysis from documents in a corpus to the words within the corpus. The third part of our research (Section 4 on word embeddings) aimed at identifying some of the mechanisms which lead to the acquisition of legal meanings at the word level. We processed and trained word embeddings models on two large general language Latin corpora and compared them to models trained on the legal corpus of ROMTEXT and its sub-corpus constituted by the text of the *Digest*. We evaluated the models by a benchmark designed for general language use and adapted the benchmark to assess how well these word embeddings models capture legal meaning. With some necessary caveats, we were able to detect, in a data-driven way, a semantic split between the general and the legal meaning of words according to their vector representation in general and legal corpora. We expected words with dominantly general (e.g., *dubie*—"doubtfully") and words with dominantly legal meaning (e.g., *municeps*—"citizen"), respectively, to be semantically similar regardless of the context they appear in. This intuition was supported by the similarity measure we developed. The vector representation of such words and hence their semantic neighbours were largely similar in both the general and the legal corpora. Additionally, if the similarity between the representations of the same word in the general and legal corpora was low, this was found as a good indication for a semantic split between general and legal use. Even more interesting is the case of words which do not strike one as legal such as those related to family and time. An analysis of their semantic neighbours suggests that a potential shift from general to legal meaning is compatible with a view of law as practice. According to this view, a mother is not seen as a woman with a child, but as a placeholder of rights, duties and assets attached to the status a mother holds. Similarly, time is not presented as a measure of living one's life, but as the dimension by which rights, duties and assets are expressed and exercised. Consequently, the corresponding words are used in very specialised contexts in legal texts, which differ strongly from their use in general language.

We believe that the combination of "legal" and "historical" is where the novelty of our research lies. Natural Language Processing (NLP) methods have been customarily used for the analysis of legal texts as well as historical texts, but not the combination of both. As far as modern legal texts are concerned, NLP has been used, among others, for text classification [14], legal metadata [15] and rule extraction [16], automated document summaries [17] and legal question answering [18]. In a commercial context, the field has developed into a lucrative "legal tech" sector with countless services fighting for the attention of legal professionals [19]. In an academic context, the field's flagship "AI and Law" conference has clocked almost 35 years of productive existence with many specialist research communities branching out from its core [20]. Our line of inquiry is related more to the NLP research of historical texts. In the body of this paper, we refer to and rely on recent achievements which used methods of corpus linguistics for Latin texts. As much as our research stays in this latter Digital Humanities (DH) tradition, we hope that the laboratory conditions provided by Latin historical texts will produce methodological insights which are relevant for those working on modern law.

## 2. Corpus: From Raw Text to Database

Justinian's *Digest* aspired to make all previous works of Roman law redundant by preserving only the necessary and omitting everything else. This monumental historical sourcebook documents the development of Roman law over 800 years from the *XII Tables* passed by the Senate in 449 BCE [21] until the time of the jurist Hermogenianus from around 350 CE [22]. The *Digest* aimed to demonstrate antiquity, intellectual superiority, and justice manifested in Roman law. Justinian the Great (527–566 CE) wanted to provide a fresh start for an empire where the law as studied and administered fell into

a chaotic complexity. As such, the *Digest* was designed to lay the foundation for the revival of the (Byzantine) Roman Empire.

In the imperial constitution *Deo auctore* dated to 15 December 530 CE, Justinian announced "a completely full revision of the law" which should be carried out "by way of logical distinction or supplementation or in an effort toward greater completeness". The emperor appointed his chief jurist, Tribonian, to carry out the project which would create "total concord, total consistency" among the legal writers of Roman law who "will have equal weight" [23] (pp. xxxiii–xxxv). According to the imperial constitutions *Tanta* and *Dedoken*, Tribonian's committee revised "nearly two thousand books and nearly three million lines" [23] (pp. xxxvii, xliv). They selected 9,132 passages from about 300 juristic works authored by a total of 37 jurists. The committee arranged the passages in 432 thematic sections in a total of 50 books. In 533 CE, less than three years after its launch, the *Digest* project was completed.

In addition to the speed of its completion, another stunning feature of the *Digest* is that it has been preserved in a complete manuscript of extraordinary antiquity. The so-called *littera Florentina* was produced around 555 CE, shortly after the text's official publication [24] (pp. 255–256). The manuscript has been the subject of philological scrutiny since its 11th century "rediscovery" in Pisa, explained by Charles Radding and Antonio Ciaralli "as an effect of the revival of juridical culture in Italy that created an audience capable of understanding what the book had to teach" [25] (p. 10). At the climax of text-critical investigations towards the end of the 19th century, Theodor Mommsen published the complete Justinianic legal corpus [10] which remains the standard in Roman law scholarship still today. In the Enlightenment period, the *Digest* inspired reformulations by jurists such as Robert Pothier and Karl-Friedrich von Savigny. They, and Roman legal scholars in their footsteps, worked towards a system of Roman law which provided the blueprint for systematic codes and prolegomena for the study of law on the European continent [26].

This great tradition of philological and juristic study of Roman law in general, and of the *Digest* in particular, was primarily based on close reading of select passages. We offer a different corpus-based approach to uncover characteristics that are hidden from our eyes when we look at the text too closely. A computer-assisted approach may provide supplementary support for theories developed by close reading, and it may produce new conjectures worth following up with traditional methods. This is where computational linguistics has a lot to offer to the study of so-called less-resourced historical languages like Latin and of specialist knowledge domains such as that of Roman law. Methods and tools available for this language and in this domain are still few when compared to modern languages benefiting from enormous datasets, ample funding, and lucrative commercial applications. This is despite the fact that, historically speaking, scholars of Latin texts played a pioneering role in the field by adopting computer technologies for their research early on [27,28]. Computational Linguistics started in 1949 when a Jesuit priest, Roberto Busa, successfully pitched his project to IBM. By securing an unlikely funding, Busa set out to digitize the complete Latin works of St Thomas Aquinas assisted by computer-processed punch cards [29].

Even before the first volumes of the *Index Thomisticus* came to light, Marianne Meinhart proposed a similar computational project to study the Latin text of the *Digest* in 1970. The first results materialised in about 100,000 punch cards based on Mommsen's text edition and created by a dedicated research institute at the University of Linz five years later. The institute decided to make the expanded and corrected ROMTEXT database available in DOS format which enabled to run search queries in a command line interface [11]. Peter Riedlberger and Günther Rosenbaum provided ROMTEXT with a graphical user interface in the Amanuensis software [30] from which we pulled the raw text of the *Digest* for our study. Figure 1 is a screenshot from this opensource software. In this example, we search the works of the jurist Ulpian ("Ulp.") for the inflected forms of *furtum* ("theft") with a wildcard ("furtu*"). The screenshot shows how bibliographic headings and text units are arranged in ROMTEXT.

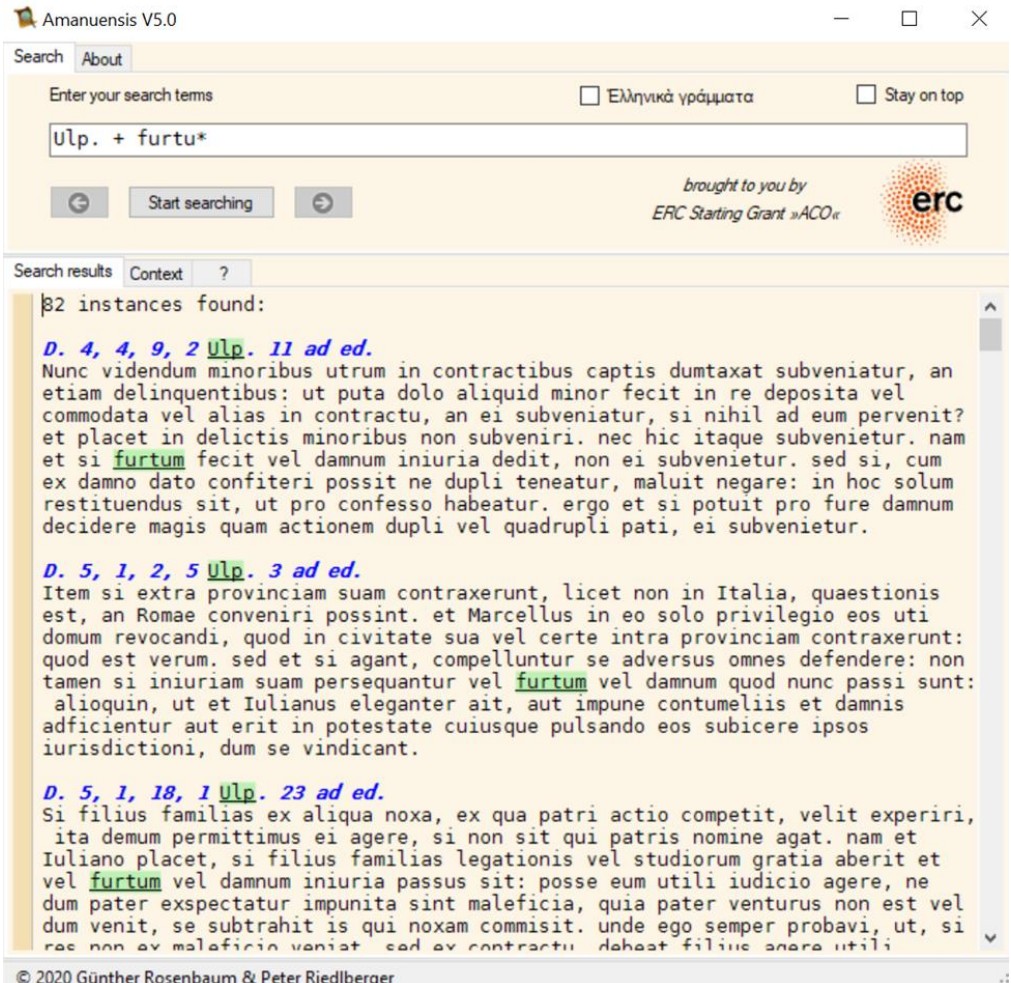

**Figure 1.** Screenshot of Amanuensis V5.0 displaying texts related to *furtum* in the works of Ulpian in ROMTEXT.

We have recently reported the first release of an SQLite relational database of Roman law based on Justinian's *Digest* [12]. While the *Digest* text in this initial release may have some typographical errors inherited from ROMTEXT, the relational database approach guarantees that information retrieval and structured quantitative analysis carried out by SQL queries provide comprehensive and fully reliable results.

The processing pipeline that led to the creation of the database included mass data transformation and alignment, followed by programmatic checks of accuracy and a manual close inspection and correction of anomalies. As the original punch cards of ROMTEXT were created by hand, some errors and inconsistencies are similar to those committed by the ancient and medieval scribes of manuscripts. For example, contributors working on the project at different times abbreviated names and titles in the inscriptions of passages and used upper case and lower case in a slightly different manner. Such errors and inconsistencies were checked and corrected according to Mommsen's *Digest*, and all changes were documented with a note and an accompanying page reference to the print edition. Processing the *Digest* text at scale has revealed some interesting philological characteristics such as messy inscriptions, peculiar editorial notes, and books and sections with an unusual structure, which may justify a separate discussion in the future. In the project's GitLab repository, we provide detailed documentation in the form of markdown files, graphs, flowcharts, and in-line comments in Python scripts to make the transformation from raw text to database fully transparent and reproducible [31]. The next section includes an example unit from the database (Ulpian, *Inst*. 1, D.1.1.6.1) illustrating the pre-processing of the text for computer-assisted analysis.

The size of the database is less than 7Mb. It includes the *Digest* corpus in 50 books, 432 thematic sections, 9132 passages and 21,055 text units, totalling 803,465 word tokens. The core database tables provide a consistent and structured presentation of the text. Thematic sections are individually titled and demonstrate strong internal cohesion to the point that Friedrich Bluhme suggested that their creation followed a strict editorial formula [32]. Bluhme's compositional theory and its controversial enhancement by Tony Honoré [33,34] are the sources of information of an additional table. The database enables to date individual passages and text units by linking them to entries in the supplementary "jurist" table which gives the estimated date of the *Digest*'s 37 jurists based on biographies in reference works [35–37]. The reuse potential of the database is discussed in detail in a dedicated data paper [12]. Our purpose was to provide a solid foundation for large scale quantitative analyses of the *Digest* corpus such as the one demonstrated in this paper. We also wanted to assist future corpus-based projects which either focus on Roman law or study the complexity of the ancient world in a linked data approach where legal texts are used alongside other sources.

## 3. Clustering Analysis: Discovering the Empirical Structure of Roman Law

For our clustering analysis, we rely on the valuable ancient "metadata" of the 432 sections to create thematic clusters, extract keywords, and eventually build an empirical structure of Roman law. Even though we take sections as the basic unit of our investigation, we do not aim to confirm or challenge the compositional theory by Bluhme and Honoré [32,33]. We only adopt the following two minimal assumptions from their line of inquiry. First, we assume that sections are indeed thematically coherent, and, second, we assume that the 432 sections provide a comprehensive picture of what Roman law is. As the titles of the sections themselves grant us a very limited amount of textual data, we only use them for control purposes. The basis for clustering and keyword extraction is the text of the passages these sections include.

Our analysis is designed to bring out a conceptual structure from the empirical evidence of the *Digest*'s text. As such, it needs to be distinguished from the effort of detecting inferential structure of statements and rules arranged in a systematic order. In the context of the Archimedes Project of the Department of the Classics at Harvard University and the Max Planck Institute for the History of Science in Berlin, researchers were investigating the history of mechanics and engineering from antiquity to the Renaissance period. Demonstrating the benefits of a computational approach, Harvard's Mark J. Schiefsky presented a method to build a graph representation of inferential relationships between propositions in Euclid's *Elements* [8]. Schiefsky focused on logical connectives which are used systematically in the work. Our goal is different from his inasmuch as we do not assume an inferential structure in the *Digest*. We focus on content words rather than logical connectives. We refer here to Manfred Fuhrmann's research on the ancient "study books" of scientific inquiry which underlined the terminological nature of technical language [38]. We aim to achieve a structured presentation of key terms and concepts of Roman legal science without assuming an architectonic system of legal rules.

The 50 books of the *Digest* are relatively insignificant for the purpose of identifying an underlying thematic structure. These books are practical units of production and dissemination, and they are thematically coherent only by virtue of the sections they include. The role of the book in the *Digest* is consistent with ancient practice one may notice in the Bible and other ancient Jewish works [39], as well as in the texts of the Greek-Latin speaking Mediterranean [40]. According to David Pugsley, Tribonian's editorial team also took the book as the unit of production which made it possible to publish parts before the project has been fully completed. Pugsley suggests that the books of the *Digest* were released in stages to the law schools of the Empire which were able to adopt the material as it originated from school practice in the first place [13].

The starting point of our analysis consisted of 21,055 text units organized into 432 thematic sections of the *Digest* pulled from raw text files (.csv and .txt) based on the database described in Section 2. A series of pre-processing steps were necessary in order to apply clustering algorithms to the text. Figure 2 summarises the steps on the example of a mixed Greek and Latin sentence from

Ulpian's *Institutes* 1 preserved in D.1.1.6.1. We removed punctuation, superfluous white spaces, and the characters of occasional Greek quotations (non-ASCII). We split the text on white space to create a list of word tokens, and then, we generated the dictionary form (lemma) of each inflected word. For the lemmatization of inflected Latin word tokens, we used the BackoffLatinLemmatizer (BLL) developed by Patrick J. Burns [41] for the Classical Language Toolkit (cltk), a Python-based NLP framework for classical languages inspired by the Natural Language Toolkit (nltk) [42]. Burns' BLL combines multiple lemmatizer tools in a backoff chain passing the word token to the next tool in the chain until a lemma is returned, or the chain runs out of options. It should be noted that lemmatization was found to be incomplete as the BLL was not able to identify the lemmas of some inflected forms.

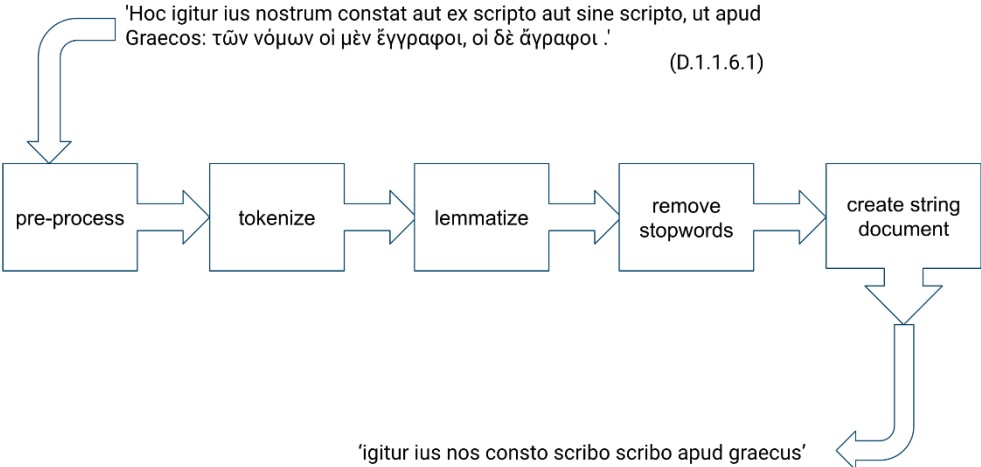

**Figure 2.** Pre-processing and the creation of lemmatized texts on the example of a passage from the *Digest* (Ulpian, *Inst*. 1, D.1.1.61).

In order to focus on content words, we removed common words like *sum* ("is"), *qui* ("who/which"), and *is* ("he") which are prevalent in all sections but not characteristic of their vocabulary. From a corpus perspective, these stopwords with high frequency and low semantic value obscure rather than contribute to the linguistic character of a given section. For example, in sections related to the use of public property, we wanted to give weight to words such as *servitus* ("servitude"), *aqua* ("water"), *via* ("road"), and *fundus* ("land") without the noise created by words which do not contribute to the substance of the section. Acknowledging that general lists of stopwords may not be appropriate to a specialist corpus of legal Latin, we decided to create a custom list based on the lemmatized text. We imported cltk's Latin Stop module [43] and used its highly customizable "build_stoplist" method to create a list of high frequency words. We manually downselected the list to create custom stoplist of 68 words available in the "D_stoplist_001.txt" file in the project's GitLab repository [31]. The stoplist includes pronouns (e.g., *quis* and *hic*), conjunctives (e.g., *et* and *sed*) and verbs (e.g., *habeo* and *facio*) among others.

We created bag-of-words ("bow") versions of the documents from the pre-processed, tokenized and lemmatized text. A bow document includes the lemmas of text units in a given thematic section as a list which disregards the linear order of words in the original sentence. The assumption is that the content of the bow document characterises the section without complicating matters by recording the order of words. The 432 bow documents produced in this step correspond to the 432 thematic sections.

In order to cluster the thematic sections as our units of inquiry into groups, we needed a way to represent each unit as a vector, that is, a sequence of numbers which can be visualized as a point in a geometrical space. We could then apply different geometric measures to identify the points that are closest and group them together. We used Term Frequency-Inverse Document Frequency (tfidf) to vectorize the bow documents created from the lemmatized text of the 432 sections. We imported the "TfidfVectorizer" function from the scikit-learn Python package to calculate scores for terms in

a "document" which forms part of a "corpus". The tfidf score of a term in a document indicates its "importance" in the document relative to the term's overall "importance" in the corpus. The score is calculated as the dot product of term t's term frequency (tf) and its logarithmically scaled inverse document frequency (idf) where tf is the number of times term t appears in a document divided by the total number of terms in that document, and where idf is the natural logarithm of the total number of documents divided by the number of documents with term t in it. For example, the lemma *furtum* ("theft") appears a total of 787 times in the *Digest* corpus, including 396 times (50.32%) in the section *de furtis* ("Thefts", D.47.2). The occurrences of *furtum* are unevenly distributed among the sections of the corpus, which is exactly what the tfidf score captures. The score ranges between 0 and 1, so the very high tfidf score of *furtum* (0.8250) in *de furtis* indicates the lemma's high relative importance in that section. While most sections do not use *furtum* at all, few of them use it a lot, and it is fair to assume that the topic of *furtum* plays a significant role in them. We treat such high-scoring lemmas as the keywords of the section, and when such keywords are taken as a set, we can make an educated guess about the theme of the section without reading the passages it includes.

The bow documents created from the lemmatized text of the thematic sections were passed to scikit-learn's TfidfVectorizer. The returned matrix has 432 rows, corresponding to the number of documents, and 10,865 columns, corresponding to the number of unique lemmas or "features" in the corpus. The values of the cells in the matrix are the tfidf scores of each lemma in the text of each of the 432 thematic sections. We are dealing with extremely sparse and high dimensional data stored in the tfidf matrix. Each section is described by 10,865 features corresponding to the number of unique terms in the corpus, but most of these terms do not appear in the text of a given section and hence their value there is zero. The average number of unique lemmas in a section is 347.34 which means that the average number of zero values in a section is 10,517. In order to improve the quality of clustering, we took two additional steps to reduce sparsity and dimensionality of data in the tfidf matrix. First, we downsampled the 432 sections by removing 93 which have fewer than 100 unique lemmas. Second, we restricted the tfidf matrix to include common terms only. We selected the 50 lemmas with the highest tfidf score in at least one of the 339 sections that remain after downsampling. These steps resulted in a leaner tfidf matrix in the shape of 339 × 4029 corresponding to 339 sections described by 4029 features.

We performed hierarchical clustering on the leaner matrix with the scipy Python package to create a linkage matrix which can be visualized in a tree-like dendrogram as shown in Figure 3. The figure is only presented here to give an idea about the dendrogram's shape, and we refer to the full resolution image "norm_top50_ward_euc_clusters.png" in the project's GitLab repository [31] for a closer analysis. Hierarchical clustering creates clusters from the bottom-up by iteratively merging pairs of items with the most similar features. Features for this "new" merged item are recalculated based on the features of the constituent items. Merging and recalculation of features are repeated until all units are collapsed into one at what becomes the trunk of the tree. From the many options available, we selected the clustering algorithm which uses Ward's method with Euclidean distance [44]. This method-metric pair produces larger clusters at lower distances. It creates a stronger inner-cluster cohesion at the expense of deviating more from the original units when they are merged together. On the left-hand side of the tree, units representing the 339 sections are separated on the vertical axis. The horizontal axis measures the distance between units. This is used to ascertain at what distance two particular units are merged into one during the clustering process. As the distance increases along the horizontal axis, more and more units are merged from leaves to twigs, to branches, and finally to the dendrogram's trunk.

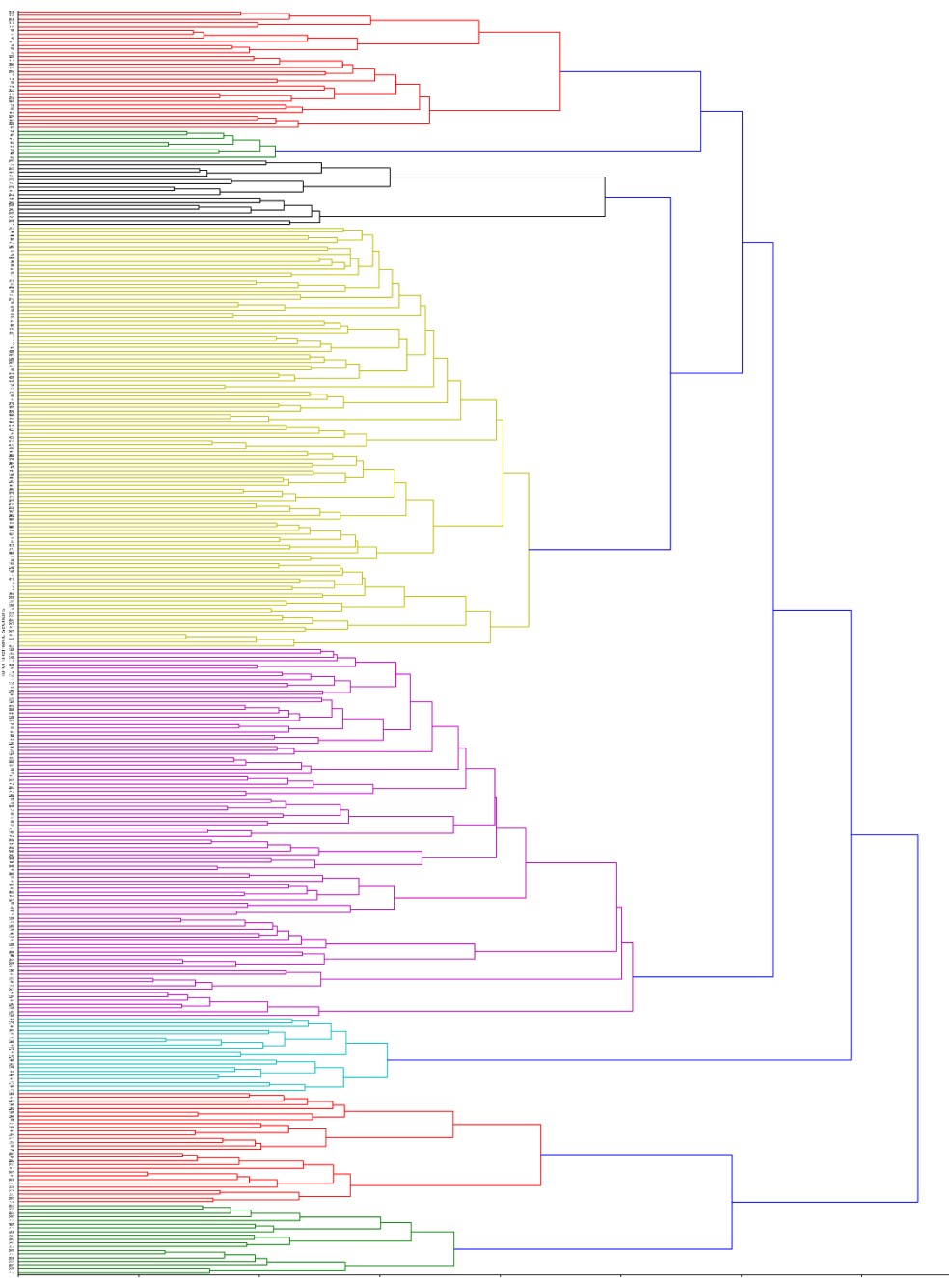

**Figure 3.** Dendrogram based on hierarchical clustering with Ward's method and Euclidean distance performed on the tfidf matrix of thematic sections of the *Digest*.

Ward's method produces an intuitive result on data which is still extremely sparse and characterised by high dimensionality. This is generally the case when the corpus is small like ours and includes semantically rich textual data. We experimented with standard K-means clustering and used the silhouette score measure to pick the ideal number of clusters between 2 and 75. For all numbers, silhouette scores stayed at an abnormally low level, suggesting that clustering with K-means produces a very unreliable result. The reason for this behaviour is that the mean-based algorithm at the heart of K-means is notoriously sensitive to outliers and unfit for processing high-dimensional sparse data [45]. These characteristics of our data also explain why hierarchical clustering algorithms with other method-metric pairs were unable to produce meaningful results. Initially, we defined a function to calculate the cophenetic correlation coefficient (CCC) for all method-metric pairs to select what

we though would be the optimal combination. The idea is that the closer the CCC score is to 1, the closer features of the created clusters stay to the features of the original units [46]. The highest scoring method-metric pair of "average" and "minkowski" [47] with 0.7128 merges units at high distances producing a high number of small clusters. Other method-metric pairs with a high CCC-score produced similar results. We realised that high dimensional sparse data with a large number of zeros push vectors to the edges of the vector space, which results in largely dissimilar units that most method-metric pairs are unable to cluster efficiently. We decided to use Ward's method despite its low CCC-score because it is capable of producing meaningful clustering on high-dimensional sparse data.

We transformed the dendrogram created with Ward's method to a conceptual tree-map of Roman law as shown in Figure 4, which can be accessed for closer analysis as "clast_graph.png" in the project's GitLab repository [31]. The tree-map arises empirically from the vector representation of the texts of thematic sections. Each cluster box shows the number of constituent thematic units. Additionally, the box lists the top ten keywords of the cluster to indicate a possible common theme as shown in two examples in Figure 5. The graph was built manually in the yEd graph editor based on a machine-assisted inspection of the content of clusters in the dendrogram and its underlying linkage matrix. We made wide cuts closer to the trunk of the tree and finer cuts towards the twigs and leaves so that each cut produces approximately twice as many clusters as the previous one. To extract keywords, we created documents from thematic sections in the same cluster and generated tfidf matrices at each cut. We then recorded and displayed the ten lemmas with the highest tfidf score in each cluster. The two major branches at the largest Euclidean distance are divided into five, ten, seventeen, thirty-one, fifty-five, and eighty smaller branches, twigs, and leaves in subsequent cuts made at smaller distances. We horizontally aligned clusters which are formed at the same Euclidean distance. When the cluster was unchanged between cuts, we displayed it only at the lower distance to avoid cluttering the graph.

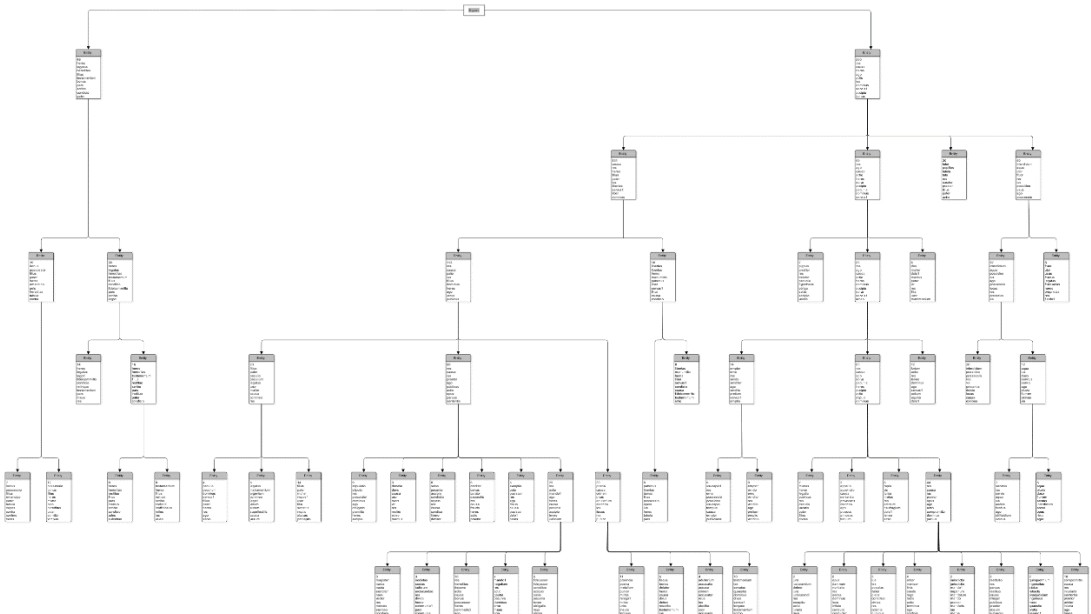

**Figure 4.** Empirical conceptual tree-map of Roman law based on thematic sections of the *Digest*.

```
┌─────────────────────┐        ┌─────────────────────┐
│    "manumission"    │        │      "theft"        │
├─────────────────────┤        ├─────────────────────┤
│ 8                   │        │ 12                  │
│ libertas            │        │ furtum              │
│ manumitto           │        │ actio               │
│ heres               │        │ res                 │
│ liber               │        │ teneo               │
│ servus1             │        │ dominus             │
│ condicio            │        │ ago                 │
│ causa               │        │ servus1             │
│ fideicommitto       │        │ actium              │
│ testamentum         │        │ aquilia             │
│ ratio               │        │ dolo1               │
└─────────────────────┘        └─────────────────────┘
```

**Figure 5.** Two example clusters from the empirical conceptual tree-map of Roman law.

A close inspection of the tree-map suggests that branches, twigs, and leaves of thematic clusters are formed according to a practical consideration. We take a closer look at two example clusters presented in Figure 5 which are located at the fourth level of the tree-map formed by cutting the dendrogram at a Euclidean distance of 2.0. It should be noted that as tfidf scores and the corresponding keywords of clusters are recalculated at each cut, keywords in a cluster need to be interpreted relative to other keyword sets produced at the same cut. The one which contains twelve sections is characterised by keywords like *furtum* ("theft"), *actio* ("legal action"), and *servus* ("slave"). Such list indicates a common theme of "theft" which is confirmed by section titles in the cluster like *de dolo malo* ("Malice or fraud", D.4.3), *de furtis* ("Thefts", D.47.2), and *de servo corrupto* ("The action for making a slave worse", D.11.3). Similarly, the cluster with eight sections and keywords like *libertas* ("freedom"), *manumitto* ("emancipate"), and *testamentum* ("will") suggests a common theme of "manumission". Sections in this cluster have titles like *de bonis libertorum* ("Freedmen's property", D.38.2), *de obsequiis parentibus et patronis praestandis* ("The obedience to be offered to parents and patrons", D.37.15), and *de adsignandis libertis* ("The assignment of freedmen", D.38.4) which indeed relate to "manumission".

Relying on the similarity of the tfidf vector representation of sections allows us to identify thematic affinity between sections where the ancient titles are ambiguous. For example, *de actione rerum amotarum* ("The action for property unlawfully removed", D.25.2), in which the jurists discuss the breakup of households and the dispersion of property between their members, would not be readily associated with "theft", if one relies on the title alone. As the jurists in this section argue, a common way for recovering property in such circumstances is to bring a case in theft or fraud which justifiably puts the section in a cluster related to "theft". Similarly, while the title *quarum rerum actio non datur* ("In which cases an action is not given", D.44.5) gives away little about the topic, the passages included in this section are indeed related to "manumission" as it is primarily about preventing freedmen from bringing an action against their former masters. With a modicum of caution, tfidf cluster assignment can be taken as a good indication of thematic affinity, and in certain cases, the assignment uncovers hidden relations between sections found far away from each other in the *Digest* corpus.

We have so far only assumed internal coherence of sections for the purpose of forming thematic clusters. It is time to take advantage of our other assumption which holds that the 432 sections of the *Digest* are not only individually well-formed, but they also constitute a comprehensive representation of what Roman law is. That is the key for producing the tree-map and inspecting the common themes of clusters in the context of the whole corpus. Figure 6 is a simplified conceptual tree-map of Roman law created on the basis of the clusters and keywords displayed in Figure 4. While generating keywords in a cluster is based on the empirical measure of tfidf scores, it should be noted that translating them into a common theme is based on subjective interpretation which could (and should) be contested.

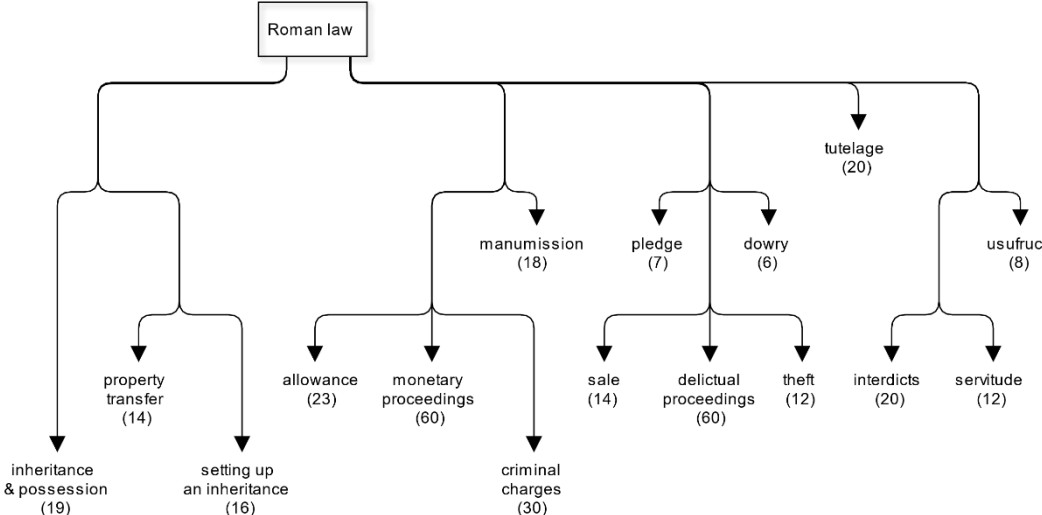

**Figure 6.** A simplified empirical conceptual tree-map of Roman law.

As we noted above, we performed hierarchical clustering on 339 sections which have at least 100 unique lemmas. Figure 6 is a simplified empirical tree-map of Roman law in which we identified common themes in larger clusters and noted the number of sections they include. The tree-map suggests two main branches of Roman law. The smaller one with 49 sections (14.45%) relates to the topic of inheritance and divides into smaller themes about possession, property transfer, and the setting up of an inheritance. The larger branch with 290 sections (85.55%) includes a strong cluster related to tutelage which forms at an early stage and three additional branches of law. Clusters related to usufruct, servitude, and interdicts form a branch which could be associated with the broad concept of property use. The other two branches seem to be less homogeneous. One of them is dominated by proceedings related to monetary issues with smaller clusters about manumission, criminal charges, and the allowance. The other larger branch is dominated by proceedings related to delicts and torts among others with smaller clusters about pledge, dowry, sale, and theft.

The empirical tree-map we have here is nothing like the neat structures of law created by jurists with a theoretical or philosophical inclination. The so-called institutional framework of law in Justinian's *Institutes* divides law into public and private parts. The private part divides to law derived from sources of nature, nations, and states, and those derived from states divide further to the law of persons, things, and actions (*Inst.* 1.1). The *Institutes* is based on the work with the same title by the 2nd century CE jurist Gaius [48]. This work further distinguishes between corporeal and incorporeal things (Gaius *Inst.* 2.12-14), a distinction which "has sparked ardent discussions amongst Romanists and private lawyers alike" as Francesco Giglio puts it [49] (p. 127). The tree-map in Figure 7 illustrates the institutional framework which we compare and contrast to our empirical tree-map.

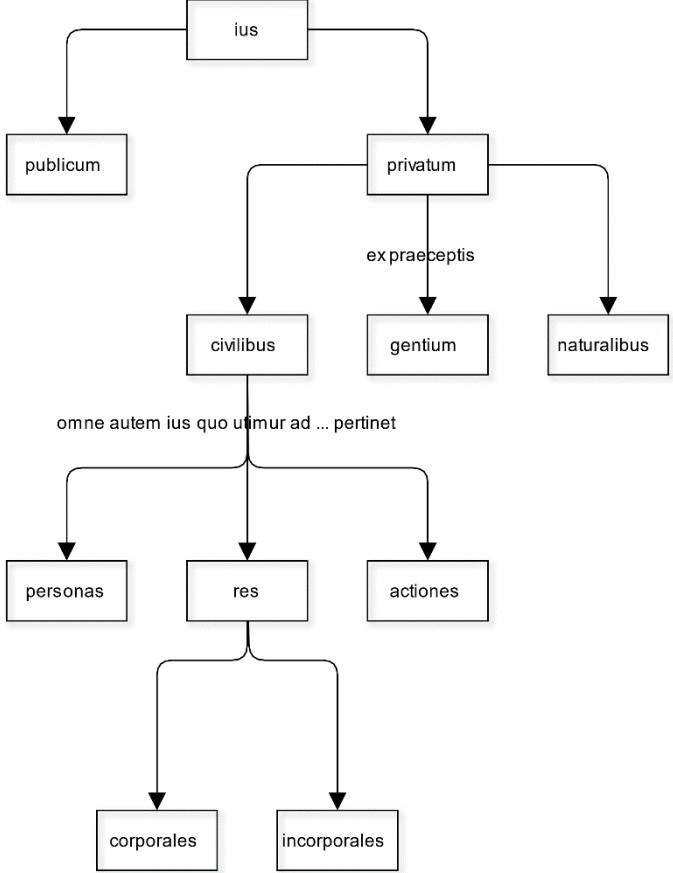

**Figure 7.** The institutional structure of law according to Justinian's *Institutes* 1.1 (533 CE) and the *Institutes* of Gaius 2.12–14 (165 CE).

Giglio points to the rich scholarship about Greek philosophical influence on Roman law, while putting Gaius' distinction between corporeal and incorporeal in the context of Stoic and Sceptic philosophy. However, Greek philosophy and Roman law proved to be somewhat incompatible with each other. According to René Brouwer, later Stoics were unhappy with the philosophical foundation (or the lack thereof) of key Roman legal concepts such as "justice" and "equity" [50], while the jurists, according to Alberto Burdese, were immune to philosophical accuracy as their main interest was in the law as it is practiced [51]. For these reasons, the institutional framework had little impact on the educational practice of ancient law schools, which seem to have ignored it in favour of a hitherto unknown principle.

Scholars suggested that the codification ideal for Justinian's project was the *Perpetual Edict* [52] and for this reason, the distribution of topics in the *Digest* shows some affinity with it [53]. David Pugsley went further and suggested that the *Digest* could be edited, circulated and adopted so quickly because it relied on pre-existing practice of law schools which developed their own textbooks and curricula primarily based on the commentaries of the jurist Ulpian [13]. Tribonian's editorial committee seem to have made choices coherent with contemporary legal education, which focused on legal practice rather than philosophical neatness.

We do not claim that this practical focus is inherent to Roman law itself. We would rather distinguish between two meanings of "Roman law": Roman law as a set of texts, and Roman law as an intellectual construction. The latter one can be reconstructed as an architectonic inferential structure in a similar way to what the 19th century Pandectists achieved. The two structures of Roman law, the empirical and the intellectual one, are not rivals in the sense that if one is true, then the other is necessarily false. They present Roman law from different perspectives and for different purposes. Our goal in this paper is to demonstrate that there is an underlying conceptual structure in

the empirical textual evidence of Roman law based on Justinian's *Digest*, a structure which suggests that the organising principle of law is a practical one.

## 4. Identifying Semantic Splits between Legal and General Vocabulary Using Word Embeddings

Our empirical investigation of Roman law has suggested a practical focus at the basis of its structure. As we now zoom in on the lexical-semantic properties of legal terms, we argue that this characteristic also manifests itself at the level of words. Our aim was to capture semantic split in words which display both a general meaning and a specialized legal meaning, and how this split can be detected via automatic methods. We developed a method based on word embeddings which identifies words deviating from their general meaning in legal texts. By extracting the semantic neighbours of a given word, we were also able to illustrate the semantic contexts of use which distinguishes a word's general and legal meaning.

We used the evaluation benchmark for measuring performance of word embeddings models trained on general Latin corpora developed by the Linked Latin (LiLa) project [54]. In order to measure the performance of models trained on legal texts, we supplemented the LiLa benchmark with our own by creating a subset of words with semantic associations pulled from Adolf Berger's *Dictionary of Roman law* [55], which we call "legal gold standard set". Our semantic similarity score, which compares the semantic neighbours of a given word in general and legal language corpora, provides a good indication for this semantic split.

Following the so-called distributional hypothesis, semantically similar words share similar contexts of use [56]. For a few decades, computational linguistics have employed vector representations of the textual contexts where a word is found in order to model its semantic properties [57]. At the most basic level, the method counts how many times words appear in the vicinity of the target word in a given corpus. With this simple frequency-based method, the context words become the features of the target word and the number of times they appear in the target word's context become the number associated with that feature. The method allows to create n-dimensional vectors for each word in the corpus where n is the number of words appearing in the corpus. The geometric proximity between word vectors in this n-dimensional space can be interpreted as semantic relatedness. These methods have been successfully applied in digital classics studies, and particularly archaic Greek epic [58].

Dense low-dimensional vector representations deploying neural network algorithms such as Google's word2vec [59] and Facebook's fastText [60] have recently been found to effectively model certain aspects of lexical semantics. Such static word type embeddings algorithms have been shown to be outperformed by Bidirectional Encoder Representations for Transformers (BERT) token embeddings in a range of state-of-the-art NLP tasks, including semantic similarity analysis [61]. However, in the first shared task on unsupervised lexical semantic change detection [62], token embeddings showed worse results compared with static type embeddings. For this reason, the specific purpose of grasping the semantics of the legal genre suited fastText better. Moreover, the BERT algorithm requires input data at a larger scale than we have available for a genre group like legal writings. Our aim was not simply to find the best general word sense model, but to approximate and illustrate a possible semantic split between general and legal language use. For this purpose, we opted for static word embeddings algorithms to capture semantic differences when the input data is relatively small. David Bamman and Patrick J. Burns report about promising results in building BERT for Latin [63]. An investigation whether BERT could be used for capturing genre-dependent semantic differences in a small corpus like ours would be the subject of future research.

Rachele Sprugnoli and her colleagues report on experiments of training and evaluating Latin lemma embeddings with word2vec and fastTexts algorithms [54]. The authors test these methods on two large annotated Latin corpora. One of them is the Opera Latina corpus of classical Latin authors created and maintained by the Laboratoire d'Analyse Statistique des Langues Anciennes (Lasla) at the University of Liège [64] which includes 154 works from 19 authors totalling 1,630,825 words [65]. The other corpus is Roberto Busa's Corpus Thomisticum including the annotated works of Thomas

Aquinas [66]. Their evaluation shows that fastText outperforms word2vec, the skipgram method (skip) method outperforms continuous bag-of-words method (cbow), while performance hardly increases (if at all) when using word vectors with higher dimensions, that is, 300 instead of 100. The reason for fastText's superiority may be that word2vec is word-based, while fastText performs on character level which usually works better for highly inflected languages like Latin. We suspect that there is a similar explanation for the finding that skipgram outperforms the continuous-bag-of-word method.

We created fastText word embeddings models trained on two legal and two general corpora, and we adopted the optimal parameters (fastText/skip/100) for training the models from Sprugnoli [54]. Our two legal Latin text corpora are ROMTEXT and the *Digest* as its subset. We lemmatized the ROMTEXT and *Digest* corpora and excluded lemmas which appear one or two times in the corpus. Altogether, we removed 17,126 low-frequency tokens from ROMTEXT and 1636 from the *Digest* corpus. We adopted the fastText model trained on the Lasla corpus by Sprugnoli [54] which we complemented with another general Latin language model trained on Version 4 of the LatinISE corpus [67]. This freely available online corpus covers 1274 texts written between the beginnings of the Latin literary tradition (2nd century BCE) and the contemporary era (21st century CE) extracted from LacusCurtius, Intratext and Musique Deoque. Texts have been semi-automatically lemmatized and part-of-speech tagged, and the texts are enriched with metadata containing information as genre, title, century or specific date. We pulled the lemmas from the LatinISE corpus and ran cltk's BLL to lemmatize words which stayed in inflected form. Table 1 below summarises the main characteristics of the four corpora and the word embeddings models built on them.

**Table 1.** Main characteristics of the four Latin corpora and the performance scores of their fastText word embeddings models on the general and adapted legal benchmark.

| Corpus | Word Tokens | Sentences | "General" Performance | "Legal" Performance |
|---|---|---|---|---|
| LatinISE | 6,670,292 | 348,053 | 87.80% | 70.19% |
| Lasla | 1,630,825 | ~85,096 | 85.56% | 64.69% |
| ROMTEXT | 1,573,383 | 39,368 | 67.44% | 61.31% |
| Digest | 803,465 | 21,055 | 62.87% | 51.37% |

Sprugnoli and her colleagues have developed an evaluation benchmark freely available on the LiLa word embeddings website (https://embeddings.lila-erc.eu/). The benchmark includes 2756 test words coupled with a close semantic neighbour, which were checked and approved manually by a Latin expert. Test words are associated with three additional words which produce a set for a four-way TOEFL-style multiple choice question. Word embeddings model are tested on the task of selecting the right word as the closest semantic neighbour from the set of four.

We have adapted this benchmark to create an evaluation test for capturing legal meaning. We manually inspected entries in Berger's *Dictionary of Roman law* [55] and added a close semantic neighbour to 473 words to create our legal gold standard set The "syn-selection-benchmark-Latin-legal.tsv" file in the project's GitLab repository [31] includes the LiLa benchmark and the legal gold standard set as its subset. The target word appears in the first column, the Berger semantic neighbour in the second, the approved general semantic neighbour in the third. The remaining three columns include three random words which are combined with the Berger or general semantic neighbour to produce multiple-choice test questions with four options. By leaving the three additional words in the set unchanged, our subset of legal words is comparable with the general language performance which is based on the full list of 2756 words. For example, the target word *actio* is associated with the manually approved semantic neighbour *gestio* ("behaviour") in the LiLa benchmark which randomly adds *tot* ("so many"), *tyrius* ("of Tyre"), and *Oricum* (proper name) to create a multiple choice set of four.

We retain the latter three words and replace *gestio* with *iudicium* ("judgement") which appears in the dictionary entry of *actio* in Berger to create one test question in our own legal gold standard set.

We created a custom function in Python which calculates pairwise fastText similarity scores between the target word and the four possible answers in the set. If the manually approved semantic neighbour has the highest similarity score, we interpret this as a correct answer. Table 2 illustrates the evaluation on the example of the target word *actio*. The word embeddings model trained on the general LatinISE corpus produces the highest similarity score for *gestio* in the general benchmark (0.5457), and for *iudicium* in the legal benchmark (0.5840). We interpret these as correct answers. The model trained on the legal ROMTEXT corpus produces the highest similarity score for *tyrius* in the general benchmark (0.4811), and for *iudicium* in the legal benchmark (0.5763). We interpret these as one incorrect and one correct answer.

**Table 2.** Word embeddings accuracy calculated with pairwise fastText similarity scores on the example of the target word *actio*—Blue highlights the correct answers, orange the incorrect ones.

| *actio* | **LatinISE Corpus** | **ROMTEXT Corpus** |
|---|---|---|
| general benchmark | *gestio*—0.5457 | *gestio*—0.4288 |
|  | *tot*—0.0658 | *tot*—0.0669 |
|  | *tyrius*—0.0474 | *tyrius*—0.4811 |
|  | *oricum*—0.2657 | *oricum*—0.2329 |
| legal benchmark | *iudicium*—0.5840 | *iudicium*—0.5763 |
|  | *tot*—0.0658 | *tot*—0.0669 |
|  | *tyrius*—0.0474 | *tyrius*—0.4811 |
|  | *oricum*—0.2657 | *oricum*—0.2329 |

The general language evaluation uses the full set of 2756 words in the original LiLa benchmark file, while the legal language evaluation uses the 476 words in the legal gold standard set. The percentage of correct answers produced by a model is the model's evaluation score on the given benchmark. The general and legal evaluation scores shown in Table 1 rank the four corpora in the same order with LatinISE performing better then Lasla, ROMTEXT, and the *Digest* corpus. During the clustering exercise, we encountered a challenge originating from small corpus size and sparse high dimensional data (see Section 3). We believe that these characteristics explain why the smaller specialist corpora of ROMTEXT and *Digest* perform worse than the larger and general corpora of LatinISE and Lasla. The sensitivity of the word embeddings models to legal meaning needs to be interpreted in context. While performance on the legal gold standard set drops in all four corpora compared to the general gold standard, the drop is less than 10% for the legal corpora of ROMTEXT and the *Digest*, while the drop is around 20% for the general corpora of LatinISE and Lasla.

We used the words included in the general and legal benchmarks to compare the word embeddings models in more detail. For each target word, we extracted ten semantic neighbours with the highest similarity score and arranged them in structured dataframes. Lemmas in the Lasla corpus are in bare dictionary form while LatinISE marks homonymous entries with numbers according to the Latin dictionary by Lewis and Short [68]. The BackoffLatinLemmatizer in cltk which we used for ROMTEXT and Digest also generates lemmas in Lewis-Short format. In order to standardise the output and make terms comparable across the four corpora, we removed such numbers together with hyphens, underscores and capital letters which appear in the lemmas of ROMTEXT, *Digest*, and LatinISE. We saved the dataframe including semantic neighbours and similarity scores in pickle (.pkl) file format to preserve data types. We performed a pairwise comparison in the six possible combinations of the four models and created a weighted score to measure the semantic proximity of words in different corpus contexts according to the semantic neighbours they produce. The fastText similarity score ranges between 0 and 1. This characteristic lends itself to use the score itself for giving weights to semantic neighbours which appear in the top ten for a target word in two models. We took the similarity score of a word in the top ten semantic neighbours in one model and multiplied it with

the similarity score of the same word in the other model provided that the word appeared in the top ten in both models. For example, there are two overlapping semantic neighbours for the word *sacer* ("sacred") in LatinISE and ROMTEXT, namely *sacraris* ("you are consecrated"), *sacro* ("consecrate"), and *sacrarium* ("sanctuary"). Aggregating the pairwise multiplied (rounded) similarity scores of *sacraris* ($0.7696 \times 0.6633 = 0.5105$), *sacro* ($0.7461 \times 0.8097 = 0.6041$), and *sacrarium* ($0.7375 \times 0.7252 = 0.5348$) results in a score of 1.6494, which is the weighted similarity score of the word *sacer* in the LatinISE-ROMTEXT comparison. This example also shows that the automated lemmatization of the LatinISE and ROMTEXT corpora are not without faults. By calculating the mean similarity pairwise, we created a heatmap which illustrates the semantic proximity of the word embeddings models, as shown in Figure 8.

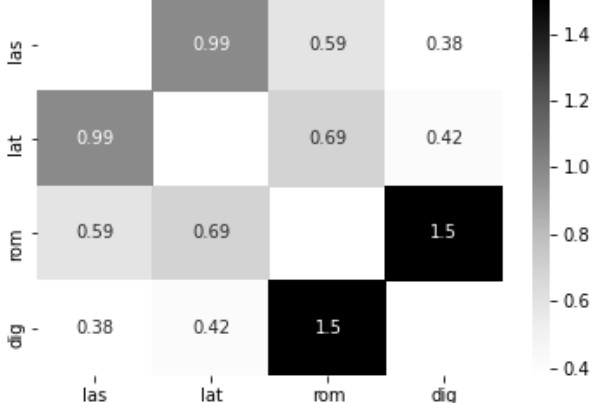

**Figure 8.** Heatmap illustrating the semantic proximity of word embeddings models based on the mean of weighted similarity scores of the top ten semantic neighbours in the four corpora (Digest, ROMTEXT, LatinISE, Lasla).

Our word embeddings models and their relative performance on the benchmarks indicated a significant semantic split between general and legal language use. There is no denying that performance scores of the models are still mostly affected by general corpus characteristics such as size and the quality of input data. With the necessary caution against drawing sweeping conclusions, the result encourages a closer look on a possible semantic split between general and legal sense at word level. Are we able to identify words which split between general and legal sense?

As expected, a high similarity score between the semantic neighbours of a word in general corpora and the semantic neighbours of the same word in legal corpora suggests that there is no semantic split. If a word is included in the legal gold standard set, then its semantics is likely to be related to the legal domain irrespective of the corpus where it is used. If it is not, then its semantics is likely to pertain to the general language. In contrast, a low similarity score between the semantic neighbours of a word in general corpora and the semantic neighbours of the same word in legal corpora is a good indication of a split in the word's semantics. An interesting group of such words are those which are not part of the legal gold standard set, and which hence were not flagged up by our manual search of a specific legal meaning in Berger's *Dictionary* [55]. The low similarity score and the different sets of semantic neighbours of inconspicuous words such as *mos* ("manner") and *amo* ("to love") indicated a previously unknown legal use. We suggest that this legal use is governed by the practical focus which also shapes the empirical thematic structure of Roman law.

We inspected words and their top 10 semantic neighbours in the word embeddings models based on LatinISE's general corpus and ROMTEXT's legal corpus to get a more nuanced understanding of semantic behaviour. For the purpose of this exercise, we have picked examples of four types of words according to two criteria. The first criterion is whether the word is included in the Berger subset of the evaluation benchmark (the legal gold standard set) and hence has a legal meaning. It should be noted here that while it is safe to treat the entries in Berger's dictionary as having a legal meaning, the fact that a word is not included in the Berger subset does not necessarily mean that it lacks legal meaning.

In order to make the comparison meaningful, we only included "legal" words which returned the correct answer in the legal evaluation test for both models, and "general" words which achieved the same in the general evaluation test. The second criterion is whether the top 10 semantic neighbours of a given word largely overlap in the general LatinISE and the legal ROMTEXT model. Here we should note the limitations resulting from a practical decision to concentrate on the top 10 semantic neighbours; semantic neighbours outside the top 10 may support a stronger or weaker affinity between the model representations of a given word in the two models. To avoid words where an empty overlap might be the result of poor model representation, we concentrated on words with at least one overlap between their semantic neighbours in the two models. The four types of words and the selected examples are summarised in Table 3. The number of overlapping semantic neighbours and their weighted pairwise similarity score are given in brackets after the word.

**Table 3.** Words selected for comparison from the evaluation benchmark and its legal subset in the LatinISE and ROMTEXT corpora; the two columns show the number of overlapping items in their top 10 semantic neighbours and the corresponding weighted similarity score shown in brackets, one indicating high similarity, the other indicating low similarity.

|  | **High Similarity** | **Low Similarity** |
|---|---|---|
| legal—words from the legal subset | *adulterium* (5/5.22) <br> *municeps* (7/4.10) <br> *fructus* (5/3.46) | *aetas* (1/0.43) <br> *tempus* (1/0.43) <br> *infans* (1/0.46) |
| general—words not in the legal subset | *consulte* (6/4.23) <br> *prehendo* (5/4.01) <br> *dubie* (6/3.56) | *mos* (1/0.38) <br> *aequus* (1/0.44) <br> *amo* (1/0.50) |

Words in the Berger subset of legal words in the evaluation benchmark ("legal" in Table 3) and with largely overlapping semantic neighbours ("high similarity") include *adulterium* ("adultery"), *municeps* ("citizen"), and *fructus* ("proceeds"). The high pairwise similarity score suggests that these words have similar vector representations in the LatinISE and ROMTEXT corpora, that is, their semantic value does not change significantly when they are used in a general or a legal corpus. The reason for such behaviour is probably that the dominant meaning of words like *adulterium*, *municeps,* and *fructus* already has a legal colour.

Overlapping semantic neighbours of *adulterium* in LatinISE and ROMTEXT include morphologically related words such as *adulter* ("adulterer") and *adultero* ("commit adultery") as well as *stuprum* ("disgrace"). It looks like that the meaning of "adultery" in the *Institutes* of Marcianus (preserved in the *Digest* 25.7.3.1) according to which "a person does not commit adultery by having a concubine" keeps its sense in Cicero's *Pro Milone* 72.3, in Vergil's *Aeneid* 6.612, or in the *Natural History* by Pliny the Younger (7.10). The same holds for *municeps* and, perhaps more surprisingly, for *fructus* where the top three semantic neighbours, *fructuosus* ("productive"), *ususfructus* ("enjoyment of proceeds"), and *fructuarius* ("usufructuary"), are the same in LatinISE and ROMTEXT. The abstract meaning of *fructus* ("the right to enjoy the profits from something") is the dominant one according to P. G. W. Glare's Oxford Latin Dictionary (OLD) [69] which also notes a concrete ("useful products"), an agricultural ("vegetable produce, crops") and a technical legal sense ("financial gain, profit, revenue"). Table 4 lists the top 10 semantic neighbours with the pairwise fastText similarity score to illustrate the output of our computational analysis. Please note that lemmatization is far from being perfect, especially the semantic neighbours returned for *municeps*. As noted above, we tried to limit this shortcoming by removing infrequent word forms and re-lemmatizing word forms pulled from LatinISE.

**Table 4.** Top 10 semantic neighbours and their fastText similarity score for three target words. Overlapping words in the top 10 semantic neighbours generated by the word embeddings models based on the LatinISE and ROMTEXT corpora are highlighted in bold. Words with an asterisk are those which are returned by the lemmatizer as homonyms with different meanings.

|  | **LatinISE** | **ROMTEXT** |
|---|---|---|
| *adulterium* | ***adulter*** (0.8979) | ***adulter*** (0.9060) |
|  | ***adulter**** (0.8865) | ***adulter**** (0.8878) |
|  | ***adultero*** (0.8778) | ***adulterinus*** (0.8824) |
|  | ***adulterinus*** (0.8512) | ***adultero*** (0.8776) |
|  | *homicidium* (0.7735) | *adulterator* (0.8189) |
|  | *incestus* (0.7651) | *adultor* (0.7851) |
|  | *impudicitia* (0.7612) | *lenocinium* (0.7175) |
|  | ***stuprum*** (0.7159) | *accuso* (0.7128) |
|  | *convinco* (0.7156) | ***stuprum*** (0.7019) |
|  | *homicida* (0.7080) | *crimen* (0.6797) |
| *municeps* | ***municipes*** (0.8079) | ***municipem*** (0.8615) |
|  | ***municipem*** (0.7727) | ***municipal*** (0.8506) |
|  | *municipis* (0.7665) | *municipalis* (0.8396) |
|  | ***municipibus*** (0.7408) | ***municipes*** (0.8173) |
|  | ***municipum*** (0.7358) | ***munia*** (0.8170) |
|  | ***municipal*** (0.6994) | ***municipum*** (0.8165) |
|  | ***municipium*** (0.6532) | ***municipibus*** (0.7862) |
|  | ***munia*** (0.6252) | ***municipium*** (0.7422) |
|  | *burdegalensis* (0.6038) | *munimen* (0.7229) |
|  | *turonicae* (0.5903) | *collegiati* (0.7085) |
| *fructus* | ***fructuosus*** (0.8658) | ***ususfructus*** (0.9004) |
|  | ***ususfructus*** (0.8641) | ***fructuosus*** (0.8416) |
|  | ***fructuarius*** (0.8634) | ***fructuarius*** (0.8233) |
|  | *usufructus* (0.8626) | *fructus** (0.8233) |
|  | *infructuosus* (0.8313) | *usus* (0.8113) |
|  | *ructus* (0.8153) | *fruor* (0.8073) |
|  | ***usufructu*** (0.8081) | *usufructuarii* (0.7945) |
|  | *fructifer* (0.8014) | ***usufructuarius*** (0.7919) |
|  | ***usufructuarius*** (0.7857) | ***usufructu*** (0.7614) |
|  | *frugis* (0.7696) | *proprietarius* (0.7556) |

A similar explanation can be provided for words with largely overlapping semantic neighbours ("high similarity") outside the Berger subset in the benchmark ("general"). Words like *consulte* ("on purpose"), *prehendo* ("take hold of"), and *dubie* ("doubtfully") are used in largely the same sense. For *prehendo* and *dubie*, the overlapping neighbours *adprehendo* ("hold on"), *reprehendo* ("hold back"), and *comprehendo* ("lay hold of"), and *dubius* ("uncertain"), *dubitatio* ("doubt"), and *indubitanter* ("without doubt") are respectively related to the main word semantically as well as morphologically. They seem to keep their dominant general sense in the legal context of the ROMTEXT corpus. For *consulte*, words which appear in the top 10 semantic neighbours in both LatinISE and ROMTEXT include *consultor* ("adviser"), *consultum* ("decision"), and *senatusconsultum* ("decision of the Senate"). Non-overlapping words in LatinISE's top 10 include *iurisconsultus* ("legal adviser") and *interregnum*, while ROMTEXT's list includes *consul* and *senatus*. Here the lists suggest a political or legal undertone for *consulte* and the words associated with it. The OLD selects examples from the works of Livy, Tacitus and Ulpian to reinforce this political legal sense [57]. The word *consulte* and its antonym *inconsulte* are used extensively in Livy's (political) history of early Rome (*Ab urbe condita* 22.38.11) and in Frontinus' work on military strategy (*Strategemata* 4.7.6). This resonates with Ulpian's use of the word in his *Opinions* (preserved in D.4.4.40.1), where a young man pays off his debt with the land belonging to his father *inconsulte* ("without due care and consideration"). As noted above, the fact that *consulte* is not included in the Berger subset of legal words does not mean that it does not have a more technical, legal meaning.

In this instance, it looks like that a political or legal sense is in fact the dominant one. The examples suggest that acting *consulte* is not simply acting "on purpose", but it is acting on advice which has been formally requested.

Words in the Berger subset of legal words ("legal") with only one overlapping semantic neighbour ("low similarity") in LatinISE and ROMTEXT include *aetas* ("age") with an overlapping neighbour *adulescentia* ("youth"), *tempus* ("time") with *dies* ("day"), and *infans* ("infant") with *impubes* ("underage"). Words which only appear in the top 10 semantic neighbours of the word in the LatinISE and ROMTEXT corpus indicate a semantic split between a general and a technical legal sense. Semantic neighbours like *pueritia* ("childhood"), *aevum* ("lifetime"), and *senectus* ("old age") in LatinISE as opposed to *fragilitas* ("frailty"), *infirmitas* ("indisposition"), and *imbecillitas* ("imbecility") in ROMTEXT indicate an interesting semantic shift. The general sense of *aetas* as a stage of life shifts to personal conditions limiting legal capacity as a result of (old) age in a legal context. The same seems to hold for *tempus* where time as perceived in one's everyday life in the units of *intermenstruum* ("new moon"), *annus* ("year"), and *semenstris* ("half year") shifts to the view of time as structuring one's legal status in the units of *decennium* ("ten years"), *vicennium* ("twenty years"), and *tricennium* ("thirty years"). Such functional semantic split in *aetas* and *tempus* is not picked up in P. G. W. Glare's *Oxford Latin Dictionary* (OLD) [69]. However, the OLD includes two separate entries for *infans*, one for "not having the power of speech" and another for "infant, little child" which reflects the (functional) semantic split indicated by the low similarity score.

Finally, we have words outside the Berger subset ("general") with only one overlapping semantic neighbour ("low similarity") such as *mos* ("manner") with *consuetudo* ("custom"), *aequus* ("even") with *aequalis* ("equal"), and *amo* ("love") with *amor* ("love"). Even though the words are not in the Berger subset, a closer look of their semantic neighbours in LatinISE and ROMTEXT suggests a semantic split. The many colours of *mos* in the general sense are reflected by its LatinISE neighbours such as *ritus* ("ceremony"), *institium* ("intercalation"), and *desuetudo* ("disuse"), but the ROMTEXT corpus associates *mos* with semantic neighbours such as *probitas* ("uprightness"), *claritas* ("splendor"), and *discordia* ("disagreement") indicating an abstract (legal) sense. Similarly, ROMTEXT neighbours such as *iniquus* ("unequal"), *debeo* ("owe"), and *indemnis* ("uninjured") bring out the legal sense of *aequus*, whereas *carus* ("dear"), *illustris* ("distinguished"), and *amoveo* ("withdraw") display a restrained side of the word *amo* which fits the general context. The OLD identifies a legal sense for *aequus* ("fair, just, reasonable, right") [69], but the legal use of *mos* and *amo* goes undetected.

Our weighted similarity calculated from the fastText similarity scores of semantic neighbours is admittedly a rudimentary one. It could be improved by considering more semantic neighbours in general and legal corpora, not just those which overlap in the top 10. It should also be noted that cltk's BackoffLatinLemmatizer have failed to return the lemma of some inflected forms. Such false lemmas adversely affected the reliability of word embeddings models and the similarity measure. It is encouraging to see that even such a rudimentary measure was able to indicate a semantic split of general and legal use despite some shortcomings in the input data. For this reason, our method promises to enrich dictionaries focusing on substantive areas of a word's meaning with the nuances of function a given word plays from context to context. In line with the fundamental idea of distributional semantics, word embeddings capture functional variations by which "age" (*aetas*) becomes a factor of limiting legal capacity and "time" (*tempus*) becomes the measure for legal rights and duties. Even when the substantive meaning is unchanged, word embeddings models can bring out the legally relevant, functional side of a word. For law, a woman with a child is a placeholder of specific rights, duties, and assets, and time is the dimension by which they are expressed and exercised. The meaning of everyday life is suspended, and the "mother" and "time" of legal texts acquire the functions they play in the legal process.

## 5. Conclusions

This article has demonstrated how advanced computer-assisted methods can generate new insights about our understanding of Roman law. We presented our quantitative data-driven approach as one complementing the qualitative tools of close reading and juristic reasoning which are dominant in traditional Roman legal scholarship. On the solid foundation of a relational database based on Justinian's *Digest*, we deployed two current computational methods to analyse the structure and vocabulary of Roman law. Hierarchical clustering of the *Digest*'s 432 thematic sections produced an empirical conceptual tree-map of Roman law. We distinguished this conceptual structure underlying the empirical textual data of Roman law as presented in the *Digest* from the inferential structure of legal rules and propositions which one may identify in Roman law as an intellectual construction. Computer assisted analysis contributes to our understanding of deep structures in Roman legal texts which may be linked to editorial choices by Tribonian's committee and the legal educational practice they presumably relied on. The inspection of the empirical conceptual tree-map suggested a practical focus which is consistent with the view that ancient law schools were interested in training lawyers for practice rather than constructing a system appealing to theorists. We should conclude that this conceptual structure complements, rather than challenges or replaces, the inherent inferential structure of Roman law as envisioned by legal theorists ancient and modern.

Our word embeddings models trained on general and legal Latin corpora indicated that this practical focus is present at level of words as well. We have developed a measure to indicate a split between a word's general and possible legal meanings. Additionally, we argued that a closer look at a word's semantic neighbours in the general and legal corpora suggest that legal meaning is "produced" by focusing on the function that the entity or action denoted by the word plays in the legal process.

Apart from substantive insights about the structure and vocabulary of Roman law, our research has potentially far-reaching methodological contributions. The SQLite relational database of the *Digest* [12] is a valuable resource in its own right, well beyond the scope of this study. It makes it possible for scholars to make structured queries and run large-scale statistical analyses on the corpus. Our research has also developed a clustering and word embeddings methodology which can be adapted to other Latin texts. Our investigation of classical and post-classical Roman law as represented by ROMTEXT could be extended, for example, to include the rich medieval and early modern reception of Roman law, and to some extent its modern adaptation as long as the language of scholarship is Latin. Alternatively, hierarchical clustering as presented in this paper could discover a hidden empirical structure in corpora which represent a knowledge domain in similar breadth as the *Digest* does for the domain of law. Similarly, our method could be adapted to grasp semantic characteristics of other technical or genre-specific usages of the Latin language via word embeddings models trained on sufficiently large corpora, and to investigate diachronic semantic change. That work could pave the way to demonstrate empirically what Mary Hesse described as the fundamentally analogical and, we should add, metaphorical nature of scientific and technical language use [70]. With sufficient data and careful selection of default values, we might be able to give a mathematical representation of the "filter" through which parts of the general language are distilled and transformed into a technical one.

**Author Contributions:** Conceptualization, M.R. and B.M.; methodology, M.R. and B.M.; software, M.R.; validation, M.R.; formal analysis, M.R.; investigation, M.R.; resources, M.R. and B.M.; data curation, M.R.; writing—original draft preparation, M.R.; writing—review and editing, M.R.; visualization, M.R.; supervision, B.M.; project administration, M.R.; funding acquisition, M.R. and B.M. All authors have read and agreed to the published version of the manuscript.

**Funding:** This research was funded by The Leverhulme Trust under the fellowship grant ECF-2019-418 and by The Alan Turing Institute under the EPSRC grant EP/N510129/1. The APC was funded by the Library Open Access Fund of the University of Surrey.

**Conflicts of Interest:** The authors declare no conflict of interest. The funders had no role in the design of the study; in the collection, analyses, or interpretation of data; in the writing of the manuscript; or in the decision to publish the results.

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
