# Peer review of "A Corpus Approach to Roman Law Based on Justinian’s Digest"

_informatics, doi:10.3390/informatics7040044_

Round 1
Reviewer 1 Report
In your opinion, are these deeper structures you have identified endemic to Roman law itself, or do these reflect the choices made by Justinian's committees? A fascinating piece, btw.
Reviewer 2 Report
This is a very interesting contribution to a general problem of how to use distributional semantics for poor-resourced languages, such as Latin. It also showcases the use of automated methods in the study of our intellectual heritage, with the potential to overthrow the received view, in this case, on the Roman law as a coherent system driven by philosophical views.
While most of the analyses are convincing, I do have several problems with the paper.
- Underdetermination
The data is so sparse that it's really difficult to draw any conclusions, in particular when one divides the Digest into 432 sections. I'm not sure if Ward's method clustering (shown in Figure 3) is anything but a statistical noise here. The hidden assumption is that semantic clusters imply the general conceptual and inferential structure of the Digest. But I hypothesize that the inferential structure, which would be the case for any philosophical or mathematical system, may not follow semantic clusters. Or at least I would think that inferential markers should be included to discover what is supposed to be implied by what.
One way to overthrow my alternative hypothesis would be to perform a similar analysis on something that is actually considered an axiomatic or philosophical system, using the same tools (possibly, the same pipeline). Euclid's Elements in the Latin translation seem a good candidate; for more modern Latin, Spinoza's Ethics would be also a suitable candidate. If it turns out that semantic clusters indeed follow the inferential / conceptual clusters that were traditionally assumed to be inherent in the given piece of text, then the analysis in the paper might have some weight. Otherwise, it could simply track the semantic clusters and nothing beyond that. The conceptual structure of the Roman law might still be highly systematic and driven by philosophical considerations.
2. Word Senses
In section 4, you compare semantic clusters and claim that you discover new word senses for several lemmas. This is interesting but the current state-of-the-art word sense induction tool is BERT+DP rather than fastText (see SemEval results). Why not run BERT on your data?
3. Hidden connections in the relational database
The conclusions section (p. 21) claims that your SQLite relational database makes it possible to "discover hidden connections between passages". How so? The paper does not give a single example, or maybe I'm missing something? All your methods rely on clear text processing (fastText or word2vec obviously are not in the business of querying SQL servers), so please elucidate this passage.
Minor problems
P. 3, line 130: misplaced space after "[15]"
p. 4, line 159: do you really mean that Busa lemmatized the works of St Thomas Aquinas? Or maybe that the purpose of the project was to digitize these works?
p. 10, line 313: spurious comma after the full stop at the end of the sentence
p. 19, Table 4: the table apparently contains inflected forms of "municeps", while on p. 14, line 447 you claim that the Digest was lemmatized for fastText. I must have missed something in between because the result suggest that either (1) lemmatization was incomplete; or (2) that fastText operated on inflected forms.
Reviewer 3 Report
Isn´t the first sentence of the abstract a bit too strict and undermining centuries of academic scholarship? Could you be a bit more diplomatic, please? Qualitative reading of ancient texts has brought a lot of ideas and contributed to European civilisation. Quantitative reading brings new insights and somehow supplement the qualitative tradition.
The English translation of the Latin sentence by Justinian is not very exact, is it? In English, it sounds a bit more like a current political correctness trend. Justinian says suum cuique tribuens.
The introduction provides the reader with a clear structure, rationale and aims. The references are adequate and sufficient.
Is it necessary to substitute BC and AD with BCE and CE? The European history scholarship sticks to the former, I guess.
Fig. 1 is not very clear to my eyes. Can you fix it, please?
Figs. 3 & 4 The captions not legible to me.
I miss some kind of discussion in your paper comparing the results of similar research in corpus linguistics and computational linguistics into legal texts. There is a lot of similar research into the topic and it could be interesting to see the novelty of your contribution in the context of previous research activity.
Conclusion:
Line 631: I don’t think you needed to demonstrate that corpus linguistics brings new insights. Everyone knows. Your contribution is that you clearly showed what the new findings are when applying the corpus linguistics approaches. The question is not IF or HOW to use it but WHAT we get when we use this tool. Can you elaborate on this a bit in the introduction and summarize it in the conclusion, please?
Line 633: it is not an alternative. It is a very different approach bringing new insights parallelly to the traditional ones.
Your references are very inconsistent and messy: bold vs not bold, hyphen vs dash, accessed on vs no information, capital letters, etc.
Again, an excellent paper, thus, my strong recommendation. I am not exaggerating to say that it has been a great pleasure to read your manuscript.
